# Effect of Prehabilitation in Form of Exercise and/or Education in Patients Undergoing Total Hip Arthroplasty on Postoperative Outcomes—A Systematic Review

**DOI:** 10.3390/medicina58060742

**Published:** 2022-05-30

**Authors:** Patrick Widmer, Peter Oesch, Stefan Bachmann

**Affiliations:** 1Klinik Für Muskuloskelettale und Internistische Rehabilitation, Rehabilitationszentrum Walenstadtberg, Kliniken Valens, 8881 Walenstadtberg, Switzerland; paediwidi@gmail.com; 2Research Department, Rehabilitationszentrum Valens, Kliniken Valens, 7317 Valens, Switzerland; peter.oesch@kliniken-valens.ch; 3Department of Geriatrics, Faculty of Medicine, Inselspital, University of Bern, 3010 Bern, Switzerland

**Keywords:** hip arthroplasty, prehabilitation, preoperative exercise, preoperative education, postoperative physical functioning

## Abstract

*Background and Objectives*: The aim of this systematic review was to determine whether prehabilitation before total hip arthroplasty, in the form of exercise therapy, education alone, or both together, improves postoperative outcomes, such as physical functioning, compared with no intervention. *Materials and Methods*: A systematic literature search was performed in the online databases PubMed, PEDro and Cochrane Library using the following search keywords: “prehabilitation”, “preoperative care”, and “total hip replacement”. *Results*: A total of 400 potentially relevant studies were identified. After title, abstract and full-text screening, 14 studies fulfilled all inclusion criteria and were included in this systematic review. Patients who completed exercise-based prehabilitation before their operation showed significant postoperative improvements compared with no intervention in the following tests: six-minute walk test, Timed Up and Go test, chair-rise test, and stair climbing. For various other assessments, such as the widely used Western Ontario and McMaster Universities Osteoarthritis Index (WOMAC), Hip disability and Osteoarthritis Outcome Score (HOOS), 36-item Short Form Survey (SF-36) and Barthel Index, no significant differences in outcomes regarding exercise therapy were reported in the included studies. Education alone had no effect on postoperative outcomes. *Conclusions*: Prehabilitation in the form of a prehabilitation exercise therapy is an effective prehabilitation measure with regard to postoperative physical functioning, while prehabilitation in the form of education has no significant effects. No negative effects of prehabilitation on the outcomes examined were reported.

## 1. Introduction

The implantation of a total hip prosthesis (also termed total hip arthroplasty; THA) is a very common surgical procedure. In Switzerland, over 20,200 primary prostheses were implanted in 2020, with both the absolute number and the annual incidence (number per 100,000 population) steadily increasing slightly since 2012 (start of data collection). Since 2012, two-thirds of patients were over 65 years old. Osteoarthritis of the hip was by far the most common indication for THA [1]. THA is considered the last treatment possibility in case of persistent pain or loss of function and failure of conservative measures [2]. Various studies have shown that it can be advantageous to perform rehabilitation before the operation, to make the patient’s condition prior to the planned operation more bearable. For example, Hermann et al. [3] showed that preoperative progressive explosive-type resistance training resulted in a significant increase in functionality and muscle strength. According to Fernandes et al. [4] a supervised neuromuscular exercise program prior to hip and knee replacement was associated with an improved quality of life pre- and postoperatively, but did not increase the overall costs of the interventions. However, it is not known whether these improved outcomes are maintained over a longer period of time postoperatively.

The aim of this systematic review was to determine whether prehabilitation brings an additional benefit in relation to the postoperative outcomes of function, pain, need for further therapies, quality of life, mental health, or medical complications, compared with no intervention. Both exercise and education were considered to be prehabilitative measures. The review included all interventions with physical training or written, visual or oral information, as well as a combination of both, that aimed to prepare patients for their forthcoming operation and for subsequent recovery and daily living.

## 2. Materials and Methods

This systematic review included controlled trials published in English examining adult patients with hip osteoarthritis and with the research question of how preoperative intervention (exercise, education) compared with no intervention prior to total hip replacement implantation affects postoperative status. Strength training, walking with assistive devices such as crutches, functional activities, mobility and cardiovascular training were considered exercise interventions. Educational interventions were defined as written, visual or verbal information for the patient in the form of teaching booklets, videos or collective multidisciplinary information sessions, which go beyond the otherwise routine preoperative information provided by the orthopaedic surgeon and anaesthetist. The pre-rehabilitative measures could be carried out in an inpatient or outpatient setting, alone at home, or under the supervision of specialist staff. All patients who received only the usual preoperative information or no information at all, and continued their usual everyday life served as control groups. All studies that compared two different preoperative interventions and/or drug interventions were excluded. Outcomes were defined as physical functioning, requirement for postoperative therapy, quality of life, pain, mental health, length of hospital stay and complications on post-surgical status.

### Search Strategy

The review was performed according the PRISMA-Guidelines. Relevant studies were searched for in the online databases PubMed, PEDro and Cochrane Library using the following search strategy: “(prehabilitat * [tw] OR prerehabilitat * [tw] OR presurg * [tw] OR pre-surg * [tw] OR preoperative care [MeSH]) AND (Arthroplasty, Replacement, Hip [MeSH] OR total hip arthroplasty * [tw] OR total hip replacemen * [tw] OR hip prosthes * [tw])”. The search was carried out in December 2021. No time interval was set for publication date. All studies published until December 2021 were eligible for inclusion in this systematic review.

The studies found were then screened independently by two raters (PW, SBa) for inclusion and, in case of disagreement, evaluated by a third rater (PO). In the first phase the studies were screened for title and abstract, followed by fulltext screening. Covidence (Covidence, Melbourne, Australia) was used as the literature management program

## 3. Results

### 3.1. Study Selection

The search strategy described above identified 400 potentially relevant studies. Based on title and abstract, 332 studies were determined to be irrelevant. A further 54 articles were excluded after fulltext screening. A final total of 14 studies were included in the systematic review. Figure 1 shows the study flow in detail.

### 3.2. Study Characteristics

All 14 included studies were controlled trials [5,6,7,8,9,10,11,12,13,14,15,16,17,18]. Ten studies were from Europe [5,8,9,11,12,13,14,15,16,17], two from Australia [7,10], one from North America [6] and one from Asia [18]. Details of the included studies are shown in Table 1. According to the Oxford Centre for Evidence-Based Medicine 2011 Levels of Evidence [19], they have all evidence levels II (Table 2).

### 3.3. Outcomes

The following seven outcomes were examined in most of the studies: physical functioning; requirement for postoperative therapy; quality of life; pain; mental health; length of hospital stay; and complications. Table 3 presents the results of the systematic review according to the intervention modality.

The prehabilitation interventions (exercise + education, exercise, education) are stratified regarding the mentioned outcomes as having a significant positive effect (+), a significant negative effect (–) or a non-significant effect (n.s.) compared with the control group. Bachmann et al. [20] yet used this approach in a former systematic review.

### 3.4. Physical Functioning

Overall, all studies apart from one [6] examined a heterogeneous number of physical functioning subgroups. Results for the outcomes Timed Up and Go test (TUG), hip range of motion (ROM), gait velocity, muscle strength, 6-min walk test (6MWT), chair-rise test (CRT), Western Ontario and McMaster Universities Osteoarthritis Index (WOMAC), and Hip disability and Osteoarthritis Outcome Score (HOOS) were reported.

13 trials provided results on physical functioning, with six showing significant improvements. Gilbey et al. [10] found an improvement in hip flexion ROM, WOMAC, 6MWT, hip muscle strength, and gait velocity. Gocen et al. [12] showed a significant difference in transfers and climbing stairs, while Holsgaard-Larsen et al. [13] reported improvements in HOOS-Sport/Rec, ascending stairs, CRT, 25 m gait and knee extension muscle strength. Oosting et al. [16] showed differences only in the CRT, Villadsen et al. [17] in hip extension and abduction, and Zeng et al. [18] in WOMAC, 6MWT, TUG and balance. Furthermore, Cavill et al. [7] showed a trend regarding improvements in the TUG, Giraudet-Le Quintrec et al. [11] regarding time until patients could stand, and McGregor et al. [15] regarding WOMAC.

### 3.5. Requirement for Postoperative Therapy

Only Butler et al. [6] were able to show how prehabilitation had an impact on the need for postoperative physiotherapy and occupational therapy. They found that the intervention group needed significantly less physiotherapy and occupational therapy (physiotherapy 7.3 vs. 9.4 sessions and occupational therapy 2.2 vs. 3.1 sessions). In addition, the absolute therapy time required was less (physiotherapy 163.8 vs. 228.2 min and occupational therapy 55.6 vs. 75.8 min). McGregor et al. [15] showed that the patients in the intervention group needed less occupational therapy, but did not provide further details. Cavill et al. [7] could not make any statement in this regard due to insufficient patient data.

### 3.6. Quality of Life

Five RCTs [5,7,8,15,17] examined the impact of prehabilitation on quality of life. Only Bitterli et al. [5] found a significant improvement in the 36-item Short Form Survey (SF-36) vitality subscore without reporting exact numbers. They reported a significant interaction effect between group and follow-up time point regarding vitality and mental health in favor of the intervention group one year after surgery. All other trials showed no significant differences in quality of life, measured either with EQ5D or percentages of persons having sleep disturbances. 

### 3.7. Pain

Ten of the RCTs [8,9,10,11,12,13,15,16,17,18] examined how reported postoperative pain differed between the two groups. Two RCTs (Ferrara et al. [9] and Giraudet-Le Quintrec et al. [11]), found a significant difference in the postoperative, Visual Analogue Scale (VAS) of 1.8 points (5.5 vs. 7.3, *p* = 0.04) (Ferrara et al. [9]) and 0.7 points (2.1 vs. 2.8, *p* = 0.04) (Giraudet-Le Quintrec [11]). Two other RCTs (McGregor et al. [15] and Zeng et al. [18]) found a trend towards a lower pain level (2.1 vs. 3.1 in the VAS or 9.3 vs. 10.6 in the WOMAC pain score).

### 3.8. Mental Health

Six included RCTs [5,6,8,9,11,15] examined mental health using different scores. Three of these described significant improvements in the intervention group compared with the control group (Bitterli et al. [5]: in the SF-36 without reporting any numbers; Butler et al. [6]: 21.6 vs. 31.2 points in the State-Trait Anxiety Inventory (STAI) (*p* = 0.007); and McGregor et al. [15]: reporting better postoperative satisfaction with surgery in the intervention group with *p* < 0.01).

### 3.9. Length of Hospital Stay

Only McGregor et al. [15] showed a statistically significant difference, with a reduction in length of stay by 3 days on average in the intervention group compared with the control group (15 days vs. 18 days).

### 3.10. Complications

The number of complications was examined in three RCTs [11,14,16]. Giraudet-Le Quintrec et al. [11] reported on complications only globally with a complication rate of 9% in the intervention group and of 6% in the controls (*p* = 0.40). Hoogeboom et al. [14] described two persons with complications in the intervention group: one person experienced a femur fracture during operation and another person suffered from a low saturation rate postoperatively. Some more details regarding complications were noticed in the study of Oosting et al. [16]. They reported on eight persons in the intervention group with complications (4 with wound complications, 1 with cardiac problems, 1 with loss of sensations, 1 with herpes zoster and 1 with orthopaedic complications). Eleven patients in the control had postoperative complications (3 wound problems, 2 cardiac problems, 2 with delirium and 1 person each with orthopaedic or renal problems, decubitus ulcers and bowel obstruction). No significant differences were found in any of the studies.

### 3.11. Sociodemographic Factors

The correlation between sociodemographic factors (gender, age and race) and postoperative outcomes after prehabilitative interventions was discussed only in two studies. Butler et al. [6] noticed that men had shorter hospital stays, lower anxiety scores and higher scores in their satisfaction of preparation for returning home, compared to women, independently of group allocation. Mc Gregor et al. [15] reported that older persons over 70 years had more helplessness (*p* < 0.05) regarding psychological measures, whereas WOMAC scores showed a trend toward a greater reduction and Barthel Index improved better in older subjects who had preoperative education (*p* < 0.05).

## 4. Discussion

### 4.1. Summary of Key Findings and Comparison with Other Studies

Based on the results of all 14 included studies, it can be stated that prehabilitation has no negative impact on any of the outcomes examined. All studies showed either a positive or no effect in the intervention group regarding postoperative physical functioning compared with the control groups without intervention.

It was notable that only one included study was from a German-speaking country [5]. This may be because no state funding is provided for prehabilitation in Switzerland and perhaps in other German speaking countries.

Although some studies did not find any significance in favour of prehabilitation with regard to post-operative outcomes, they showed that a preoperative intervention can improve functionality, strength and pain symptoms up to the time of the operation [5,9,11,16,18]. Furthermore, Butler et al. found that patients who completed prehabilitation had a lower requirement for postoperative physiotherapy and occupational therapy [6]. This was also reported by Konnyu et al. [21] in a systematic review including six RCTs with 425 patients undergoing total hip replacement. The patients in the intervention groups (exercise and education in a variety of settings and forms for two to ten weeks) required fewer sessions of postoperative physiotherapy.

Exercise, alone or combined with education, provided very similar physical functioning results. For various scores, such as WOMAC, HOOS, SF-36, Barthel Index, etc., there were many non-significant results, indicating that prehabilitation does not lead to any improvement in these scores. However, these scores may not be the appropriate assessment instruments to detect relevant differences in the patient groups examined. In actively conducted assessments, the data situation was more balanced or even more in favor of improvements. For example, all studies that used CRT, gait speed or stair climbing as functional assessments showed significant improvements. Gilbey et al. [10] found a higher gait velocity without reporting any numbers, the patients in the intervention group of Gocen et al. [12] could climb stairs earlier (after 6.2 vs. 7.4 days). Holsgaard-Larsen et al. [13] found a better CRT of 2.6 s, a higher 25 m gait speed of 1.5 m/s and stair climbing speed of 1.2 steps/s and Oosting et al. [16] a better CRT of 5.8 s. The 6MWT and TUG scores also seem to improve compared with no intervention. Cavill et al. [7] and Oosting et al. [16] did not find any significant improvement in the 6MWT or TUG scores, but both are pilot studies with only 20 or 26 patients. The outcomes ROM and muscle strength had a high heterogeneity of results, making assessment difficult.

Furthermore, regarding all physical outcomes related to functioning, it was notable that in patients who underwent prehabilitation of more than eight weeks, every outcome improved significantly. Therefore, the time factor, and thus the dose-response relationship, of the prehabilitative measures seems to play a relevant role. The dose-response relationship also applies to both outpatient and inpatient rehabilitation, which means that large effects require a high dose of therapy. The validity of the dose-effectiveness principle is strongly suspected in musculoskeletal rehabilitation, based on the Cochrane review by Khan et al. [22]. This review found evidence for the effectiveness of multidisciplinary rehabilitation after THA. Early and organized multidisciplinary rehabilitation led to more rapid functional recovery, fewer post-operative complications and shorter stay in hospital compared to usual hospital care [22]. The advantage of a multidisciplinary rehabilitation is that a higher dose of rehabilitative measures can be applied to the patients during the therapy day. Routinely, patients undergo three to five active exercise sessions performed by physio- and occupational therapists per day, leading to a therapy dose of two to three hours of therapy per day. 

With regard to mental health, a predominantly positive effect was seen both in patients who received only an exercise intervention and in those who received a combination of exercise and education. Only Ferrara et al. [9] found no significant difference using the SF-36 three months after surgery. An explanation for this could be the fact that the patients in both groups got the same postoperative inpatient rehabilitation for one month. McGregor et al. [15] showed that patients in their intervention group had lower expectations regarding the effect of the operation, and these expectations were then exceeded postoperatively. Overall, patients felt they were well prepared for what was to come postoperatively [16]. Exercise alone or in combination with education does not appear to affect postoperative pain, quality of life, length of hospital stay, or complications of any type.

A purely educational intervention did not have any postoperative influence on physical functioning, pain, quality of life, mental health, length of hospital stay, or complications. Although Clode-Baker et al. [8] found no benefit in terms of postoperative mental health, the patients were satisfied with the increased information they received before the stay.

A Cochrane Review by McDonald et al. [23] analysed 15 RCTs with 1074 patients undergoing total hip replacement to determine whether preoperative education improves postoperative outcomes with respect to pain, function, health-related quality of life, anxiety, length of hospital stay and complications (e.g., deep vein thrombosis and infections). The patients in the intervention groups of the included trials got verbal, written and audio-visual education for up to six weeks. This Cochrane review described in the intervention group, compared with the control group, a 26% better relative WOMAC score improvement postoperatively (mean scores by 4.84 points lower in the education groups, 95% CI −10.23 to 0.66), less days needed postoperatively to standing or walking (mean difference 0.12 days), a shorter length of hospital stay (mean difference 0.79 days), pretty identical ROM of hip flexion and abduction, a 11% reduction in various pain scores (mean VAS with usual care 3.1 (0–10), 0.34 points lower with preoperative education (95% CI −0.94 to 0.26), a 7% reduction in anxiety measured with the State-Trait Anxiety Inventory (mean postoperative score with usual care 32.16 on a 60-point scale, 2.28 points lower with preoperative education (95% CI −5.68 to 1.12) and a 21% reduction in complications (infections, thrombosis and other serious events). However, all these improvements were statistically non-significant. The authors concluded therefore that in people undergoing hip replacement, preoperative education may not offer additional benefits over usual care [23].

Although sociodemographic factors (e.g., age, gender) might play a role regarding outcomes after prehabilitative interventions, it is notable that only two studies addressed this shortly [6,15], showing that male gender and older age can be factors influencing the results. Based on our systematic review we are not able to conclude which sociodemographic factors should be addressed in order to improve postoperative results in patients undergoing prehabilitation.

Our systematic review included only studies that compared an intervention group with a control group without any intervention. In contrast, Rooks et al. [24] compared preoperative exercise vs. preoperative education. They found that the exercise intervention group did not differ in the function subscores of the assessments WOMAC and SF-36, in pain and muscle strength compared with the education group eight and 26 weeks postoperatively. This would be consistent with our findings, as we also found no study reporting a benefit in either intervention regarding these outcomes. However, surprisingly, Rooks et al. [24] also found no significant difference in the TUG. Nevertheless, more complications occurred in the exercise group. Furthermore, treatment in the exercise group was associated with a higher probability of being discharged home postoperatively without rehabilitation.

Gill et al. [25] compared land-based vs. pool-based exercise for people waiting for THA or total knee arthroplasty (TKA). They found no significant differences in the TUG or the CRT. Based on the findings of the current study it can be assumed that, although exercise training is useful, it is not primarily a question of whether the sessions are performed in the water or in a gym. However, Gill et al. [25] reported less pain immediately after the water sessions compared with the land-based training in a gym. Doiron-Cadrin et al. [26] compare a tele-prehabilitation program and an in-person prehabilitation program with usual care over twelve weeks for THA candidates. The preoperative data have been published to date, and no differences could be found either in the WOMAC and SF-36 scores or in the TUG and stair test assessments. The postoperative data are not yet published.

The current systematic review only provided data from patients expecting a total hip replacement. Irrespective of the prehabilitative modality, there were no effects of the interventions with regard to length of hospital stay and complications. A more far-reaching and very comprehensive meta-analysis of 178 RCTs was provided in 2021 by Perry et al. [27], who examined the influence of prehabilitation prior to major elective surgeries (such as orthopaedic, cardiac, or abdominal surgical procedures). They identified eight types of preoperative interventions: nutritional, respiratory, exercise, multimodal, educational, psychological, smoking and alcohol cessation, and pharmacological. It was noted that exercise prehabilitation (physiotherapy, cardiotraining, strengthening) had no effect on the length of hospital stay, but had a positive effect on postoperative pulmonary complications (Risk Ratio RR 0.54, *p* = 0.0003). Likewise, education (booklets, videos or telephone calls) had no additional effect on length of hospital stay. Patients with combined interventions of 2–4 different modalities, all of which included exercise, went home on average 1.7 days earlier (*p* < 0.00001) and had fewer complications (wound infections, pulmonary complications, pneumonia; RR 0.84, *p* = 0.02) than patients with usual care.

### 4.2. Strengths and Limitations

This study has a number of strengths. The systematic review included studies from four different continents and studies with only level of evidence II. This provides a good global picture, which is particularly stringent from a scientific point of view. Although the included works examined a wide range of interventions and outcomes, they could be divided relatively well into the three relevant classes: exercise, education and a combination of these interventions. Thus it was possible to evaluate and present a variety of prehabilitative measures.

A limitation of this study is that due to the heterogeneity of the reported outcomes, it was not possible to carry out a meta-analysis and to evaluate the results statistically using odd ratios. In addition, some of the included randomized controlled trials had a small number of patients.

### 4.3. Further Research

The effect of prehabilitation including future studies from a wider number of countries should be investigated further, since many of the included studies were from English-speaking or Scandinavian countries. Also, various factors like socioeconomic status, gender, race, etc. could potentially influence prehabilitation and should be addressed further. Future investigations should include a cost-benefit analysis. Furthermore, a four-arm study could provide further insights that directly examine the effect of exercise vs. education vs. exercise and education vs. no intervention.

## 5. Conclusions

In summary, prehabilitation in the form of exercise was an effective prehabilitation measure with regard to postoperative physical functioning concerning actively conducted assessments like chair rise test, gait speed or stair climbing. Various other scores, such as WOMAC, HOOS, SF-36, and Barthel Index may not be the appropriate assessment instruments to detect relevant functional differences in the patient groups examined. Prehabilitation in the form of education alone had no significant effect regarding postoperative functioning, indicating that this prehabilitative intervention offers no additional benefit over usual care. Both, exercise and education seem to have no or only a small effects on postoperative quality of life, pain, length of hospital stay and complications. None of the included studies showed a negative influence of prehabilitation on the outcomes examined.

## Figures and Tables

**Figure 1 medicina-58-00742-f001:**
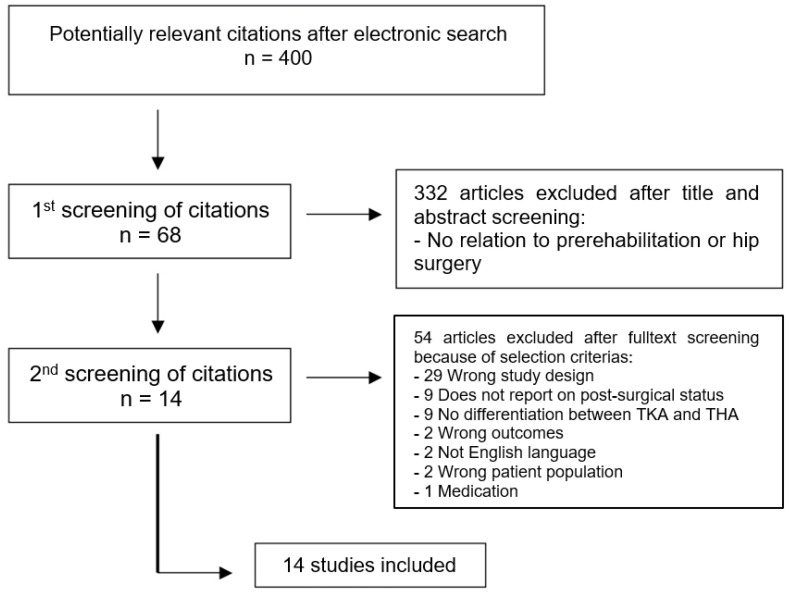
Study flowchart. TKA—total knee arthroplasty, THA—total hip arthroplasty.

**Table 1 medicina-58-00742-t001:** Characteristics of included studies.

Study & Design	Reference	Country	Level of Evidence	Population	Intervention Group	Main Outcome Measures	Positive Significant Outcomes Regarding Intervention Group
Bitterli et al., 2011 RCT	[5]	Switzerland	Level II	80 patients with arthrosis or femoral head necrosis undergoing first and unilateral total hip endoprostheses	Pre-surgical sensorimotor training programme at home over 2–6 weeks	Physical functioning (Balance, SF-36, WOMAC)	Higher mental health and quality of life (SF-36: no numbers reported) 12 months after surgery
Butler et al., 1996 RCT	[6]	Canada	Level II	80 patients undergoing first total hip replacement; age range 17–85 years	Teaching booklet with multidisciplinary inputs 4–6 weeks before surgery	Anxiety score (STAI)	Less anxiety at discharge (STAI: 21.6 vs. 31.2); Lower requirement for postoperative physiotherapy and occupational therapy (physiotherapy 7.3 vs. 9.4 sessions and 163.8 vs. 228.2 min and occupational therapy 2.2 vs. 3.1 sessions and 55.6 vs. 75.8 min)
Cavill et al., 2016 RCT	[7]	Australia	Level II	20 patients undergoing elective hip arthroplasty with RAPT-Score > 5	Prehabilitation (exercise and education) in a community rehabilitation centre twice weekly and at home for 3–5 weeks before and up to 6 weeks after surgery	TUG, quality of life (EQ-5D-3L) and PSFS	None
Clode-Baker et al., 1997 RCT	[8]	UK	Level II	78 patients undergoing elective total hip replacement	Education in form of a video, a booklet and plastic models 4 weeks before surgery	Hip function (NHP), HADS, Stress Arousal Checklist (SAS), pain (descriptive ordinal scale), sleep disturbance, satisfaction	None
Ferrara et al., 2008 RCT	[9]	Italy	Level II	21 patients with primary osteoarthritis undergoing first unilateral total hip replacement	Physiotherapy programme (strength, flexibility, cardiovascular training for 5 days/week) and education for 1 month before surgery	Muscle strength of hip abductors and quadriceps, ROM hip abduction and external rotation, disability (WOMAC, SF-36), quality of life, impairment, pain (VAS)	Lower pain (VAS 0.3 vs. 1.3)
Gilbey et al., 2003 RCT	[10]	Australia	Level II	67 patients undergoing total hip replacement	Exercise (aerobic, strength, mobility, cardiovascular) twice weekly clinic-based and twice weekly home-based for 8 weeks before and up to 12 or 24 weeks after surgery	Thigh strength, active hip ROM, gait function, pain/stiffness/function (WOMAC and Harris Hip Score), 6MWT	Higher hip strength 12 and 24 weeks after surgery; improvement in WOMAC total score (and domains of stiffness and physical function) 3, 12 and 24 weeks after surgery; higher gait velocity 3, 12 and 24 weeks after; better hip flexion ROM 12 and 24 weeks after; better 6MWT score 24 weeks after; no numbers reported
Giraudet-Le Quintrec et al., 2003 RCT	[11]	France	Level II	99 patients with primary osteoarthritis undergoing first unilateral total hip replacement	Collective multidisciplinary information session and a leaflet 2–6 weeks before surgery	Anxiety score (STAI), first day of walking after surgery, length of hospital stay, satisfaction	Less pain after surgery (VAS-10)
Gocen et al., 2004 RCT	[12]	Turkey	Level II	59 patients with osteoarthritis undergoing total hip replacement	Exercise (strengthening, stretching) 3 times daily and education for 8 weeks before surgery	Harris Hip Score, ROM hip abduction, pain, transfers, climbing stairs	Earlier transfers (bed 2.9 vs. 3.3 days, toilet 4.2 vs. 5.1 days and chair 4.2 vs. 5.6 days) and climbing stairs (after 6.2 vs. 7.4 days)
Holsgaard-Larsen et al., 2020 RCT	[13]	Denmark	Level II	80 patients with primary osteoarthritis undergoing total hip replacement; age > 50 years	Progressive resistance training for 10 weeks before surgery	HOOS, ascending stairs, chair-rise test CRT, 25 m gait, muscle strength	After 3 months: better improvements in HOOS-Sport/Rec (10.5 more), ascending stairs (1.2 steps/s more), CRT (2.6 s less), gait 25 m (+1.5 m/s), knee extension muscle strength (both sides), after 12 months: ascending/descending stairs (+1.3 and +1.6 steps/s)
Hoogeboom et al., 2010 RCT	[14]	Netherlands	Level II	20 elderly patients (Clinical Frailty Scale >1) with osteoarthritis undergoing total hip replacement, age > 70 years	Exercise (walk, leg press, ergometer, functional training) at least twice weekly (supervised and at home) for 3–6 weeks before surgery	Feasibility of intervention (adverse events, adherence etc.), HOOS, time to reach functional independence	None
McGregor et al., 2004 RCT	[15]	UK	Level II	35 patients with primary osteoarthritis undergoing first unilateral total hip replacement, age range 51–92 years	Booklet with further information, exercises etc. (also discussed in a hip class) for 2–4 weeks before surgery	Function (WOMAC, Harris Hip Score, Barthel Index), pain (VAS)	Shorter length of hospital stay and reduced cost of procedure (15 vs. 18 days), higher Satisfaction
Oosting et al., 2012 RCT	[16]	Netherlands	Level II	26 frail patients (Identification of Seniors at Risk (ISAR) >1) with osteoarthritis undergoing total hip replacement, age > 65 years	Exercise (functional activities and walking capacity) twice a week supervised and additionally 4 times weekly on their own for 3–6 weeks before surgery	Functional activity (CRT, TUG, 6MWT, HOOS), adverse events, length of stay	Better CRT (–5.8 s) 6 weeks after surgery
Villadsen et al., 2014 RCT	[17]	Denmark	Level II	84 patients with osteoarthritis undergoing unilateral total hip replacement, age >17 years	Supervised neuromuscular exercise programme (aerobic, core control, postural orientation, lower extremity muscle strengthening, functional) twice weekly for 8 weeks before surgery	Any subscales of HOOS, ROM hip extension and abduction	Better ROM hip extension (15% more) and abduction (35% more)
Zeng et al., 2015 RCT	[18]	China	Level II	59 patients with osteoarthritis or osteonecrosis undergoing primary unilateral total hip replacement, age range 60–69 years	Home-based Tai Chi, strength and ROM training at least 5 times weekly for 12 weeks	WOMAC, 6MWT, TUG, hip ROM, single-leg stance test	Better improvements in WOMAC functional status (36.3 vs. 41.1), 6MWT (478 vs. 419 m), TUG (14.6 vs. 19.1 s), ROM abduction (31.5 vs. 28.8°)

WOMAC—Western Ontario and McMaster Universities Osteoarthritis Index, VAS—visual analogue scale, NHP—Nottingham Health Profile, STAI—State-Trait Anxiety Inventory, CRT—chair-rise test, HOOS—Hip disability and Osteoarthritis Outcome Score, 6MWT—six-minute walk test, TUG—Timed Up and Go test, ROM—range of motion, RCT—randomized controlled trial, SF-36—36-item Short Form Survey, OR—odds ratio, EQ-5D—EuroQol health questionnaire, PSFS—Patient-Specific Functional Scale, RAPT–Risk Assessment and Prediction Tool, HADS—Hospital Anxiety and Depression Scale.

**Table 2 medicina-58-00742-t002:** Oxford Centre for Evidence-Based Medicine 2011 Levels of Evidence [19].

Levels of Evidence	Description
Level I	Systematic review of randomized trials or *n*-of-1 trials
Level II	Randomized trial or observational study with dramatic effect
Level III	Non-randomized controlled cohort/follow-up study
Level IV	Case-series, case-control studies, or historically controlled studies
Level V	Mechanism-based reasoning

**Table 3 medicina-58-00742-t003:** Results.

	Exercise + Education			Exercise			Education		
	+	−	n.s.	+	−	n.s.	+	−	n.s.
Physical functioning	[12]: Transfers, climbing stairs		[7,9,12]: Walking, ROM hip abduction, [15]	[10,13,16]: CRT, [17]: ROM, [18]		[5,14,16]: TUG, 6MWT, HOOS, [17]: HOOS			[8,11]
Requirement for postoperative therapy							[6]		
Quality of life			[7,15]	[5]		[17]			[8]
Pain	[9]		[12,15]			[10,13,16,17,18]	[11]		[8]
Mental health	[15]		[9]	[5]			[6]		[8,11]
Length of hospital stay	[15]		[7,12]			[14,16]			[6,8,11]
Complications						[14,16]			[11]

n.s.—non-significant, CRT—chair-rise test, HOOS–Hip disability and Osteoarthritis Outcome Score, 6MWT—six-minute walk test, TUG—Timed Up and Go test, ROM—range of motion.

## Data Availability

For original data reported in our systematic review we refer to the original publications.

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
