# Peer review of "Effect of Prehabilitation in Form of Exercise and/or Education in Patients Undergoing Total Hip Arthroplasty on Postoperative Outcomes—A Systematic Review"

_medicina, 2022, doi:10.3390/medicina58060742_

Round 1

Reviewer 1 Report

I would like to congratulate the team for the effort they made preparing the manuscript and providing the scientific community with a broad perspective on how prehabilitation (both exercise and education) could influence final outcomes in total hip replacement.  

In our facility, more than 80% of surgical interventions involve joint replacements. Rehabilitation and planning a cut-edge final outcome is absolutely mandatory nowadays and the topic should always be underlined. However, the manuscript require additional information and could be enriched as described below.  

First of all, the title could be misleading, especially for orthopedic surgeons. "Hip surgery" could involve many other types of procedures. Please replace “hip surgery” in the title with a more suitable term (eg. hip replacement, hip arthroplasty, total hip arthroplasty, total hip replacement - I will leave this to the authors choice).  

Row 16 - consider replacing “categories" with “keywords”;

Row 30 - consider removing the keyword “outcome” or replace it with something that is not too general;

Row 38 and further, Results section - Even some of the issues were addressed in the Discussion section, I think adding a more detailed phrase on what quality of life, pain, mental health, length of hospital stay, and complications improvement on prehabilitation protocols had would add value to the manuscript. Please enumerate and elaborate especially on the complications section;

Row 69 - add “.” period at the end of the sentence;

Row 86 - I think elaborating more on what "significant improvements” could add quantitative value to the phrase;

Row 96-96 - consider elaborating on this phrase as it is of extreme importance;

Row 100 - can we obtain a clear phrase on the hypothetical reason why Ferarra did not report any differences? Can you explain, maybe, the methodology used by their team?;

Row 112-113 - Can we add a describing sentence on the Cochrane analysis? Consider being more granular in respect to what that study aimed, and then proceed with their results (already added by your team);

Row 120-121 - readers might be interested on at least few of the statistical results; especially being the case of a big Cochrane analysis;

Row 239 - please fix the reference mention 19 and 20 - something is wrong at on that row;   

Besides the aforementioned changes, a moderate english and style revision is advised. An article that provides such systematic analysis should have a smooth reading flow process. Also, adding additional 1000-1500 characters phrases in the Results section is advised; try to be more granular in terms of description of each of the analyzed variables. This could include a broad description of the end-result from the studies included.   I suggest that the Conclusion section should include a wider conclusion-based approach on different outcomes that the study assessed; A four sentence phrase as a conclusion to a systematic review would be discouraging for our readers. From my perspective, too many references are older than 5-7 years. If this could be improved, it would add novelty to your manuscript. However, being a systematic review, this could be accepted.

Author Response

Point to point answer to reviewer 1 comments

We would like to thank the reviewer for his valuable inputs. They were very helpful for the revision process and helped us to increase the quality of our manuscript.

Below we answer point-by-point what changes were made in the manuscript:

I would like to congratulate the team for the effort they made preparing the manuscript and providing the scientific community with a broad perspective on how prehabilitation (both exercise and education) could influence final outcomes in total hip replacement.  In our facility, more than 80% of surgical interventions involve joint replacements. Rehabilitation and planning a cut-edge final outcome is absolutely mandatory nowadays and the topic should always be underlined. However, the manuscript require additional information and could be enriched as described below.  

First of all, the title could be misleading, especially for orthopedic surgeons. "Hip surgery" could involve many other types of procedures. Please replace “hip surgery” in the title with a more suitable term (eg. hip replacement, hip arthroplasty, total hip arthroplasty, total hip replacement - I will leave this to the authors choice).  

Hip surgery replaced by “total hip arthroplasty”.

Row 16 - consider replacing “categories" with “keywords”;

Categories replaced by “keywords” as proposed

Row 30 - consider removing the keyword “outcome” or replace it with something that is not too general;

Outcome replaced by “postoperative physical functioning”

Row 38 and further, Results section - Even some of the issues were addressed in the Discussion section, I think adding a more detailed phrase on what quality of life, pain, mental health, length of hospital stay, and complications improvement on prehabilitation protocols had would add value to the manuscript. Please enumerate and elaborate especially on the complications section;

Done. Details from the different studies are added at each paragraph. Regarding complication rate we also report now in more details

Row 69 - add “.” period at the end of the sentence;

Done

Row 86 - I think elaborating more on what "significant improvements” could add quantitative value to the phrase;

More details of the different studies are added.

Row 96-96 - consider elaborating on this phrase as it is of extreme importance;

Additional information regarding this review is added.

Row 100 - can we obtain a clear phrase on the hypothetical reason why Ferarra did not report any differences? Can you explain, maybe, the methodology used by their team?;

A hypothetical reason is added. We believe, that there is a correlation with the same postop program patients received.

Row 112-113 - Can we add a describing sentence on the Cochrane analysis? Consider being more granular in respect to what that study aimed, and then proceed with their results (already added by your team);

Done. The study aim is added and more details of the review are shown.

Row 120-121 - readers might be interested on at least few of the statistical results; especially being the case of a big Cochrane analysis;

Some statistical results are added.

Row 239 - please fix the reference mention 19 and 20 - something is wrong at on that row;   

Corrected

Besides the aforementioned changes, a moderate english and style revision is advised. An article that provides such systematic analysis should have a smooth reading flow process. Also, adding additional 1000-1500 characters phrases in the Results section is advised; try to be more granular in terms of description of each of the analyzed variables. This could include a broad description of the end-result from the studies included.  

Done. Details of studies and results are included

I suggest that the Conclusion section should include a wider conclusion-based approach on different outcomes that the study assessed; A four sentence phrase as a conclusion to a systematic review would be discouraging for our readers.

Conclusion expanded

From my perspective, too many references are older than 5-7 years. If this could be improved, it would add novelty to your manuscript. However, being a systematic review, this could be accepted.

Our search was done in December 2021. All published studies until December 2021 were eligible for inclusion. Our search revealed only one recent study published in 2020, all other were unfortunately “older”. Therefore, we are not able to improve this situation

Reviewer 2 Report

In this study, Widmer P and colleagues conducted a systematic review of various studies published in literature, which evaluated correlations between exercise or education or combination of both strategies used prior to hip surgery and patient’s post-operative outcomes. Of the 400 studies identified, 14 studies fulfilled the inclusion criteria defined by the authors. Various parameters like six-minute walk test, Barthel index, etc. were assessed. Overall, the analysis revealed that prehabilitation involving exercise therapy was indeed helpful with patient outcomes post-surgery but education prior to surgery did not show significant effects. The authors have done a great job at compiling the data and the review in well written. The table provides a succinct description of the various reported studies along with their major findings. Overall, the manuscript is well written. The authors are recommended to include some details about various factors that could potentially influence prehabilitation e.g., socioeconomic status, gender, race, etc. in the discussion section. Also, can the authors address if there was any bias which resulted in education not having a significant impact on post-operative outcomes?

Author Response

Reviewer 2

Point-to-point responses

We thank the reviewer for his valuable inputs. They were very helpful for the revision process and helped us to increase the quality of our manuscript.

Below we answer point-by-point what changes were made in the manuscript:

In this study, Widmer P and colleagues conducted a systematic review of various studies published in literature, which evaluated correlations between exercise or education or combination of both strategies used prior to hip surgery and patient’s post-operative outcomes. Of the 400 studies identified, 14 studies fulfilled the inclusion criteria defined by the authors. Various parameters like six-minute walk test, Barthel index, etc. were assessed. Overall, the analysis revealed that prehabilitation involving exercise therapy was indeed helpful with patient outcomes post-surgery but education prior to surgery did not show significant effects. The authors have done a great job at compiling the data and the review is well written. The table provides a succinct description of the various reported studies along with their major findings. Overall, the manuscript is well written.

The authors are recommended to include some details about various factors that could potentially influence prehabilitation e.g., socioeconomic status, gender, race, etc. in the discussion section.

We included a short paragraph in the result section, in the discussion and the impact for further research.

 Also, can the authors address if there was any bias which resulted in education not having a significant impact on post-operative outcomes?

We didn’t find any bias in the studies that could explain why education had now effect. The same was evident in the Cochrane review by McDonald. Therefore, an explanation would be speculative and therefore we did not address this in the discussion.

Round 2

Reviewer 1 Report

Requirements were fulfilled by authors and the manuscript should now be ready for publication. The issue regarding references no longer requires fixes.

Author Response

Dear reviewer

Thank you for your positive response to our revision of the manuscript. As I have seen no further tasks are needed.

Once more many thanks for your helpful comments for the firts revision process.

Sincerly

Stefan Bachmann